# Choice between implants in knee replacement: protocol for a Bayesian network meta-analysis, analysis of joint registries and economic decision model to determine the effectiveness and cost-effectiveness of knee implants for NHS patients—The KNee Implant Prostheses Study (KNIPS)

Elsa M R Marques ![ORCID],[1] Jane Dennis ![ORCID],[1] Andrew D Beswick ![ORCID],[1] Julian Higgins,[2,3] Howard Thom,[2,3] Nicky Welton,[2,3] Amanda Burston,[1] Linda Hunt,[1] Michael R Whitehouse ![ORCID],[1,3] Ashley W Blom[1,3]

For numbered affiliations see end of article.

**Correspondence to**
Andrew D Beswick;
andy.beswick@bristol.ac.uk

## ABSTRACT

**Introduction** Knee replacements are highly successful for many people, but if a knee replacement fails, revision surgery is generally required. Surgeons and patients may choose from a range of implant components and combinations that make up knee replacement constructs, all with potential implications for how long a knee replacement will last. To inform surgeon and patient decisions, a comprehensive synthesis of data from randomised controlled trials is needed to evaluate the effects of different knee replacement implants on overall construct survival. Due to limited follow-up in trials, joint registry analyses are also needed to assess the long-term survival of constructs. Finally, economic modelling can identify cost-effective knee replacement constructs for different patient groups.

**Methods and analysis** In this protocol, we describe systematic reviews and network meta-analyses to synthesise evidence on the effectiveness of knee replacement constructs used in total and unicompartmental knee replacement and analyses of two national joint registries to assess long-term outcomes. Knee replacement constructs are defined by bearing materials and mobility, constraint, fixation and patella resurfacing. For men and women in different age groups, we will compare the lifetime cost-effectiveness of knee replacement constructs.

**Ethics and dissemination** Systematic reviews are secondary analyses of published data with no ethical approval required. We will design a common joint registry analysis plan and provide registry representatives with information for submission to research or ethics committees. The project has been assessed by the National Health Service (NHS) REC committee and does not require ethical review.

### Strengths and limitations of this study

► Bringing evidence from multiple randomised controlled trials (RCTs) together in systematic reviews and meta-analysis with thorough assessments of risk of bias and confidence in the evidence ensures that health professionals have the information they need to deliver a high-quality patient experience with safe, clinically effective and cost-effective treatments.

► If there are insufficient data to build the economic model using the network of RCT studies in the literature, we will base our economic model on analyses of registry data. The joint registries that we will analyse include large numbers of patients with information on knee replacement constructs used and patient demographics such as age and sex to enable adjustment for potential confounders.

► A limitation of the systematic review elements of our project is the likely inclusion of small RCTs with short follow-up; many RCTs of knee replacement constructs report radiographic outcomes, which do not necessarily require large sample sizes and long-term follow-up. Some studies we identify may not report relevant outcomes or be at high risk of bias that will reduce the number of studies and knee replacement constructs that we can include in analyses.

► A limitation of our registry analyses is that treatment choices are made for individual patients, and uncontrolled confounding factors may influence outcomes.

► Exploring the effectiveness of knee replacement constructs in both RCTs and registries, and linking this information to implant and healthcare costs, will allow us to compare the lifetime cost-effectiveness of different knee replacement constructs.

Study findings will be disseminated to clinicians, researchers and administrators through open access articles, presentations and websites. Specific UK-based groups will be informed of results including National Institute for Health Research and National Institute for Health and Care Excellence, as well as international orthopaedic associations and charities. Effective dissemination to patients will be guided by our patient–public involvement group and include written lay summaries and infographics.

**PROSPERO registration number** CRD42019134059 and CRD42019138015.

## INTRODUCTION

In 2018, more than 100 000 primary knee replacements were performed in the UK[1 2] and over 15 400 in Sweden, our collaborating country.[3] About 96% of operations are to treat end-stage osteoarthritis, with the majority in women (57%) at a mean age of 69 years.[1] For many people, knee replacements are highly successful and can last a person's lifetime.[4] However, when a knee replacement fails, most commonly due to loosening or wear, people may require revision surgery. Revision surgery is more complex for clinicians to perform, is difficult for patients to recover from, is associated with further complications and need for rerevisions and is costly.

### Description of the intervention

The knee joint consists of three compartments: the medial femorotibial, the lateral femorotibial and the patellofemoral, all of which can be affected by osteoarthritis with associated pain and disability. Depending on which compartments are affected, a surgeon may perform a total knee replacement (TKR) or a unicompartmental knee replacement (UKR). In TKR, both the medial and lateral compartments are replaced as well as the trochlea (groove) on the front of the femur. The patella that articulates with this may be resurfaced or not. Some surgeons favour UKR on the basis of radiographic evidence of osteoarthritis affecting a single medial or lateral compartment with estimates of patient eligibility for UKR as high as 48% of all people receiving knee replacement,[5] although actual rates of utilisation are around 10%.[1 3] Compared with TKR, UKR surgery requires a less invasive procedure, retains more bone and native ligaments, the operation has a shorter duration, is quicker for patients to recover from and is cheaper for the NHS. However, with time, osteoarthritis can develop in the other compartments, and a patient with a UKR is more than twice as likely to undergo revision than a patient who has had a TKR.[1] Revision of UKR to TKR can be complex and may include treatment of bone defects.[6] Thus, many surgeons favour TKR as the first treatment for severe knee osteoarthritis. In the National Joint Registry for England, Wales, Northern Ireland and the Isle of Man (NJR), 87.4% of knee replacements performed in 2018 were TKR and 11% UKR.[1] In 2018, in the Swedish Knee Arthroplasty Register (SKAR), 90.0% of knee replacements were TKR and 9.6% UKR.[3]

In both TKR and UKR, the remaining cartilage and some bone in the affected portion of the joint is removed and replaced, typically with implants made of metal fixed to the bone and polyethylene bearing surfaces between the metal implants or affixed directly to the bone. Stability of the knee relies to a large degree on the ligaments in and around the knee. In UKR, these are retained; in TKR, some of them are removed and their function compensated for by the implants used.

### TKR construct options

Surgeons and patients may choose from a range of implant components and combinations that make up knee replacement constructs, all with potential implications for how long a knee replacement will last.

In a modular TKR construct, the polyethylene liner between the metal femoral and tibial components can be 'fixed' to the tibial component or 'mobile' with movement of the liner permitted on the tibial component. In a 'monobloc' implant, the top of the tibia is replaced with a one-piece polyethylene or metal backed implant that is fixed directly to the bone.

TKR constructs can accommodate keeping the posterior cruciate ligament (cruciate retaining) or not (cruciate sacrificing or posterior stabilised). Depending on the condition of the patella and the opinion of the surgeon, the patella can be resurfaced with a polyethylene or metal-backed polyethylene implant.

Fixation of both femoral and tibial components to their respective bones is done with or without the use of cement. Alternatively, in a hybrid TKR, the tibial component is fixed with cement while the femoral component is uncemented. More recently, inverse (or reverse) hybrid fixation with uncemented tibial and cemented femoral components has been reported as a distinct combination.[7] Patella resurfacing can be with a cemented or cementless design (table 1).

### UKR construct options

Depending on which compartment is affected and replaced, UKRs can be medial UKR, lateral UKR or patellofemoral UKR. As with TKR, component options are available to surgeons performing a UKR. The polyethylene liner may be fixed or mobile, and implants can be fixed to bone with cement, without cement (uncemented) or using a combination (hybrid). In patellofemoral UKR, the back of the patella and trochlea of the femur are resurfaced with polyethylene and metal, respectively. In rare situations, surgeons may perform multiple UKRs of different compartments at the same time[8] (table 1).

### Effectiveness and cost-effectiveness of knee replacement constructs: existing knowledge

To assess the relevance of our proposed research, we performed a scoping literature search of MEDLINE and Embase in January 2020 to identify existing network meta-analyses relating to knee replacement. Most of the 33 studies we identified considered drug treatments. Two systematic reviews and one protocol described the application of network meta-analysis to knee replacement

**Table 1** Knee construct options

| Total knee replacement | | | | |
|---|---|---|---|---|
| Bearing mobility | Fixed bearing. | Mobile bearing. | | |
| Constraint | Cruciate retaining. | Posterior stabilised. | Bicruciate stabilised. | Constrained condylar. | Hinged. |
| Fixation | Cemented. | Uncemented. | Hybrid. | Inverse hybrid. |
| Bearing materials | Metal bearing on conventional or highly cross-linked polyethylene tray. | Metal femoral component on all-polyethylene or metal tibial component (monobloc). | Other materials (ceramic bearings and ceramicised metals). | Specialised or customised implants. |
| Patella resurfacing | Patella resurfacing cemented. | Patella resurfacing uncemented. | No patella resurfacing. | |
| Unicompartmental knee replacement (medial or lateral) | | | | |
| Bearing mobility | Fixed bearings. | Mobile bearing. | | |
| Constraint | Cruciate retaining. | | | |
| Fixation | Cemented. | Uncemented. | Hybrid. | |
| Bearing materials | Metal on polyethylene. | Metal femoral component (monobloc). | | |

constructs. One study compared resurfacing of the patella with no resurfacing and, to complete the network, with patella denervation.[9] While the authors concluded that patella resurfacing was associated with a lower rate of reoperation than not resurfacing the patella and with no benefit for pain or surgeon-assessed scores, there was no attempt to interpret results in the context of assessed risk of bias of included studies. Another systematic review of different surgeries for the treatment of osteoarthritis with network meta-analysis only included cohort studies and did not assess their risk of bias.[10] A third Cochrane review protocol describing a network meta-analysis to compare diverse surgical and medical therapies in TKR[11] was withdrawn in September 2019.[12]

In a second contemporaneous search of MEDLINE and Embase for systematic reviews, we identified six that explicitly considered the cost-effectiveness of different knee replacement constructs. The potential value of patella resurfacing was evaluated in one systematic review.[13] Over 5 years, the authors concluded that there was a small cost-saving when the patella was resurfaced. However, there was no consideration of risk of bias. In one systematic review, the authors concluded that the benefit of UKR compared with TKR is dependent on the economic perspective chosen, patient characteristics and the timing of outcome.[14] In three reviews, authors concluded that UKR may be a cost-effective outcome in older patients but that this was uncertain for younger patients due to lower survivorship associated with UKR constructs.[15–17] In another systematic review comparing UKR, TKR and other procedures, results presented narratively were equivocal.[18]

A further search for economic analyses and registry studies identified 13 potentially relevant studies. Registry analyses covered bearing surfaces,[19–21] monobloc tibial components,[22] patella resurfacing,[23] mobile and high flexion designs,[20] fixation,[24] patellofemoral replacement[25] and UKR.[17 26–28] We did not find any registry studies comparable with our study comparing diverse features of knee replacement constructs. The authors of one study presented a cost-effectiveness model based on five common brands of TKR implants up to 10-year postprimary surgery.[29] Each brand studied represented a construct with cemented unconstrained components with fixed bearings, in contrast with our approach, which focuses on different constructs.

A search of PROSPERO in January 2020 identified no network meta-analyses in knee replacement comparing knee replacement constructs comparable with our planned study.

## Objectives

The objectives of our study are to synthesise evidence from randomised controlled trials (RCTs), registries and studies reporting quality of life and cost information to identify:
1. The most effective and cost-effective TKR constructs for patients of different age and sex profiles.
2. The most effective and cost-effective knee replacement constructs for patients of different age and sex profiles eligible for both TKR and UKR.

We will achieve these objectives by conducting two systematic reviews to identify RCT evidence for each question, analysis of registries from the UK and Sweden to estimate long-term revision and mortality rates, and development of a cost-effectiveness model for each question.

The primary outcome in both systematic reviews will be revision rate and timing of revision that are key markers

of construct effectiveness. Secondary outcomes will be patient-reported pain and function, and surgeon-assessed outcomes.

As the implants that make up a construct can vary in TKR and UKR, network meta-analysis with direct and indirect comparisons is an appropriate method for synthesis of data from RCTs. We will develop networks of evidence for revision (and other outcomes) at different time periods after primary TKR and UKR. These will include an initial period postprimary surgery where the risk of first revision is high, a medium-term period with a lower risk of first revision and a late revision period, where risk of first revision increases.

Recognising the short follow-up in many orthopaedic RCTs, we will also analyse data on outcomes after different knee replacement constructs from two joint registries.

Outcome data from RCTs and registries, together with evidence on quality of life and construct and health service costs, will be used to inform economic decision models that will rank and estimate the cost-effectiveness of knee constructs for patients of different sex and age profiles.

## METHODS
### Timescale
The project will collect data and perform analyses between January 2019 and December 2021.

### Patient and public involvement
This proposal was developed in collaboration with the University of Bristol Musculoskeletal Research Unit patient and public involvement group. The 'Patient Experience Partnership in Research' (PEP-R) group comprises nine members, most of whom have had joint replacement. Meetings are facilitated by a dedicated coordinator who works in partnership with researchers to provide patient and public input into research. We met with PEP-R on two occasions for input into the development of the research proposal in 2017 and 2018 and at the start of the project in 2019. Patients told us that they were not informed about knee implant options when discussing their surgery with the consultant. PEP-R will provide ongoing support throughout the study and appropriate funding is in place. Anticipated contributions will include advice on dissemination of results, particularly plain language summaries and the use of our findings to support shared decision making.[30]

### Systematic reviews and meta-analyses
The systematic reviews are registered with PROSPERO (CRD42019134059 and CRD42019138015), and a Preferred Reporting Items for Systematic Review and Meta-Analysis Protocols statement[31] is provided. The research questions are formulated according to population, intervention, control, and outcomes (PICO),[32] and review conduct will be based on methods described in the Cochrane Handbook for Systematic Reviews of

Interventions.[33] Reporting will adhere to the Preferred Reporting Items for Systematic Reviews and Meta-Analyses (PRISMA) guidelines[34] and the PRISMA extension statement for reporting of systematic reviews incorporating network meta-analyses.[35]

## Systematic review 1: TKR
### Eligibility criteria
*Types of studies*
We will include RCTs.

*Participants*
Eligible patients will be receiving elective primary TKR (unilateral or bilateral) and aged 18 years or older. There will be a diagnosis of osteoarthritis in 50% or more of the study population.

*Interventions and comparators*
We will include comparisons of knee replacement constructs as summarised in table 1. Each construct and comparator will have a combination defined by bearing mobility, constraint, fixation, bearing materials and patella resurfacing. In theory, this could be 240 different combinations, but many combinations are not feasible or desirable.

The reference construct within the network meta-analysis will be the most widely used TKR construct that has cemented, cruciate-retaining, fixed-bearing implant components with metal on polyethylene bearing materials.

*Types of outcome measures*
The primary outcome of the systematic reviews will be first revision surgery after primary TKR at any time from primary surgery. We will collect data at each time point reported in articles and, for revision outcome, extract data from published Kaplan-Meier plots if available.[36]

Secondary outcomes will include: further revision surgeries; deaths; reason for revision; patient-reported outcomes such as the Oxford Knee Score[37] and Western Ontario and McMaster Universities Arthritis Index[38]; clinician-assessed measures including the American Knee Society Score (KSS)[39] and Hospital for Special Surgery score[40]; and quality of life indices such as the EuroQol questionnaire.[41]

We will collect information on surgical complications and major adverse events including infection and deep vein thrombosis and on hospital readmissions.

### Search strategy
Online databases to be searched from inception are MEDLINE, Embase and PsycINFO on Ovid, CINAHL on EBSCOhost and the Cochrane Library. Searches of PsycINFO and CINAHL are unlikely to identify further RCTs but are routinely searched in our department. Online databases will be searched so that searches are up to date in October 2020. The search strategy for application in MEDLINE shown in box 1 will be tailored to each database. As well as RCTs, our searches will identify

**Box 1  Search strategy as applied in MEDLINE**

1. Controlled clinical trial.pt.
2. Randomized controlled trial.pt.
3. Clinical trials as topic/
4. (randomi#ed or randomi#ation or randomi#ing).ti,ab,kf.
5. (rct or "at random" or (random* adj3 (administ* or allocat* or assign* or class* or cluster or crossover or cross-over or control* or determine* or divide* or division or distribut* or expose* or fashion or number* or place* or pragmatic or quasi or recruit* or split or subsitut* or treat*))).ti,ab,kf.
6. Placebo.ab,ti,kf.
7. Trial.ti.
8. (control* adj3 group*).ab.
9. (control* and (trial or study or group*) and (waitlist* or wait* list* or ((treatment or care) adj2 usual))).ti,ab,kf.
10. ((single or double or triple or treble) adj2 (blind* or mask* or dummy)).ti,ab,kf.
11. Double-blind method/ or random allocation/ or single-blind method/
12. Or/1–11
13. (systematic or structured or evidence or trials or studies).ti. And ((review or overview or look or examination or update* or summary).ti. Or review.pt.)
14. 0266-4623(0266-4623 or 1469–493 x or 1366–5278 or 1530–440 x or 2046–4053).is.
15. Meta-analysis.pt. Or (meta-analys* or meta analys* or metaanalys* or meta synth* or meta-synth* or metasynth*).ti,ab,kf,hw.
16. ((systematic or meta) adj2 (analys* or review)).ti,kf. Or ((systematic* or quantitativ* or methodologic*) adj5 (review* or overview*)).ti,ab,kf,sh. Or (quantitativ$ adj5 synthesis$).ti,ab,kf,hw.
17. (integrative research review* or research integration).tw. Or scoping review?.ti,kf. Or (review.ti,kf,pt. And (trials as topic or studies as topic).hw.) Or (evidence adj3 review*).ti,ab,kf.
18. Review.pt. And ((medline or medlars or embase or pubmed or scisearch or psychinfo or psycinfo or psychlit or psyclit or cinahl or electronic database* or bibliographic database* or computeri#ed database* or online database* or pooling or pooled or mantel haenszel or peto or dersimonian or der simonian or fixed effect or ((hand adj2 search*) or (manual* adj2 search*))).tw,hw. Or (retraction of publication or retracted publication).pt.)
19. Or/13–18
20. Arthroplasty, replacement, knee/
21. Knee prosthesis/
22. ((arthoplast$ adj3 knee$).mp. [mp=title, abstract, original title, name of substance word, subject heading word, floating subheading word, keyword heading word, organism supplementary concept) word, protocol supplementary concept) word, rare disease supplementary concept) word, unique identifier, synonyms]
23. (knee$ adj3 replac$).mp. [mp=title, abstract, original title, name of substance word, subject heading word, floating sub-heading word, keyword heading word, organism supplementary concept) word, protocol supplementary concept) word, rare disease supplementary concept) word, unique identifier, synonyms]
24. (knee$ adj3 implant$).mp. [mp=title, abstract, original title, name of substance word, subject heading word, floating sub-heading word, keyword heading word, organism supplementary concept) word, protocol supplementary concept) word, rare disease supplementary concept) word, unique identifier, synonyms]
25. (knee$ adj3 prosthe$).mp. [mp=title, abstract, original title, name of substance word, subject heading word, floating sub-heading word, keyword heading word, organism supplementary concept)

Continued

**Box 1  Continued**

word, protocol supplementary concept) word, rare disease supplementary concept) word, unique identifier, synonyms]
26. (knee$ adj3 endoprosthe$).mp. [mp=title, abstract, original title, name of substance word, subject heading word, floating subheading word, keyword heading word, organism supplementary concept) word, protocol supplementary concept) word, rare disease supplementary concept) word, unique identifier, synonyms]
27. Unicondylar.mp.
28. Unicompartmental.mp.
29. Or/20–28
30. 12or 19
31. 29and 30

systematic reviews that will be screened for RCTs as will reference lists of relevant RCTs. Citations of key articles will be tracked in Web of Science. We will identify clinical trial records in the Cochrane Library and check them for full publication. Should no further publication be identified, we will contact authors for details of progress and study results.

No language restrictions will be applied to searches or study inclusion. Translations will be made by colleagues or professional translators when required. Studies that are unobtainable through our library (including interlibrary loans) and via author contact will be excluded. If studies have not reported any follow-up data or reporting is limited regarding outcomes and study conduct, including conference abstracts, we will contact authors for appropriate data. If not available, these studies will be excluded. A summary table of excluded RCTs will be produced.

### Data management
#### Selection of studies
Records will be imported into Endnote X9 (Clarivate Analytics). An initial screen by one reviewer will exclude clearly irrelevant articles. Subsequently, abstracts and full articles will be screened independently in Covidence by two reviewers and reasons for exclusion recorded. Discrepancies between reviewers will be resolved by consensus with involvement of an orthopaedic surgeon or methodologist. Authors will be contacted by email to confirm eligibility if necessary. Multiple reports of RCTs will be grouped together as a single study defined by a key publication with follow-up data. Notices of errata and retractions will be sought.

#### Data extraction
After piloting of forms, data will be extracted into Covidence, Microsoft Access or Excel by one reviewer. Data to be extracted will be: country; dates of recruitment; participant characteristics including indication, age and sex; inclusion and exclusion criteria; knee replacement constructs defined by bearing mobility, constraint, fixation, bearing materials and patella resurfacing; other surgical methods including approach, alignment,

computer navigation and robot assistance; rehabilitation regime; study setting to include number of surgeons and centres; sponsorship; follow-up intervals; outcome data; and information to assess risk of bias. For the categorical outcome of revision, we will collect the number of events recorded up to a particular follow-up or mean follow-up time. For continuous outcomes, we will extract means and SD or, if unavailable, make estimates based on medians, IQRs and ranges.[33] We will contact authors of eligible studies by email to maximise available detail concerning risk of bias, effect estimates and measures of variability for the outcomes of interest.

Depending on the number of studies identified, study characteristics will be fully extracted or checked against source material by a second reviewer. Outcome data will be extracted independently by two reviewers.

### Risk of bias assessment

Risk of bias of eligible RCTs will be assessed with the revised Cochrane tool (RoB 2).[42] Protocols will be sought to identify evidence of selective outcome reporting. Assessments will be made for all RCTs by two reviewers working independently with disagreements resolved with other members of the review team. Risk of bias for individual studies will be assessed as low, some concerns or high risk of bias. In surgical trials, key concerns relate to deviations from intended interventions (eg, large proportion of patients not receiving randomised allocation) and bias due to missing outcome data (eg, high or unequal loss to follow-up). Aspects of risk of bias will be considered in both meta-analysis and narrative synthesis. Data analysis in all meta-analyses will exclude studies assessed at high risk of bias.

### Data analysis
#### Pairwise meta-analysis

Data synthesis for the different knee replacement constructs defined by constraint, mobility, fixation and material will start with a tabulation of study details and narrative synthesis. The unit of analysis will be the knee.

Data will be analysed in three different time periods with the cut-offs of the period time-points varied in sensitivity analysis. Initially, the periods at risk will be: 'early stage failure' when the construct fails within the first 3 years following knee replacement; 'medium stage failure' when the construct survives the early period but fails within 10 years from the primary surgery; and 'late-stage failure' when the construct first fails 10 or more years after the primary surgery.

Data analyses will start with pairwise meta-analyses for each comparison of knee replacement constructs. The effect measure for revision will be the hazard rate ratio. Patient-reported and surgeon-assessed outcomes are likely to be continuous outcome scores, which will be analysed using difference in mean or standardised difference in mean scores.

If the pairwise meta-analyses include 10 or more RCTs, we will produce funnel plots and check for asymmetry, which may reflect publication bias.[43] If asymmetry is noted, we will perform sensitivity analyses with exclusion of small studies or conduct meta-regression.[33]

#### Network meta-analysis

We will construct a network of studies comparing different knee replacement constructs and use network meta-analysis to estimate treatment effects informed by both direct comparisons within trials and indirect comparisons across trials. HRs over the three separate time periods (early-stage, medium-stage and late-stage failure) will be assumed piecewise constant but the HR of the latter two periods will be normally distributed around that of the prior period.[44] These methods will broadly follow those we have used previously.[45 46] We do not aim to extrapolate outside of the follow-up time of RCTs as this would rely on assumptions about the parametric form; as explained further, we will use registry data for long-term rates, in line with recommendations of the National Institute for Health and Care Excellence (NICE) Decision Support Unit.[47]

Network plots will be generated for each outcome to illustrate which interventions are compared directly and indirectly and the strength of the direct and indirect evidence including the number of studies and patients available for each direct comparison. Network meta-analysis will be implemented in a Bayesian framework using OpenBUGS software (V.3.2.3).[48] We will adapt code developed for our hip replacement surgery network meta-analysis,[46] which itself was adapted from published code.[44] In our main analysis, each construct will be considered a separate intervention forming a distinct node in the network of evidence. However, this may result in a disconnected network or imprecise effect estimates. We will therefore also fit a component network meta-analysis model that assumes additivity of the components of each construct.[49] The additivity assumption will be assessed by comparing model fit between the model where each construct is distinct and the additive model.

#### Heterogeneity and subgroup analyses

We aim to fit both fixed and random effects models where possible and assess heterogeneity by inspection of the between studies SD and comparison of model fit between the fixed and random effects models.[33 50] However, we anticipate that there may be too few replications of individual comparisons to fit a random effects model, in which case we will implement an informative, evidence-based prior distribution for the heterogeneity variance.[51] Model fit is measured by the posterior mean residual deviance (which we expect to be similar to the number of data points) and the deviance information criteria[52] (which penalises fit with a measure of model complexity), with a preference for models where these measures are smaller (where differences of at least three are considered meaningful). If data allow, we will investigate whether revision rates vary according to participant, knee replacement construct and trial characteristics (including concerns

relating to risk of bias) and surgical methods including approach, alignment, computer navigation and robot assistance.

If we observe a high level of heterogeneity and cannot explain it by patient, study or surgical factors, results from the random effects meta-analysis will be reported. If no evidence of heterogeneity is found, we will report results from the fixed effect model.

We will assess consistency between direct and indirect evidence by comparing the fit of the consistency model with the fit of an inconsistency model (the unrelated mean effects model), which relaxes the consistency assumption. Model fit statistics will be compared and the contribution to the posterior mean residual deviance for each study will be plotted for the consistency model against the inconsistency model to identify any particular studies contributing to inconsistency.[50]

### Confidence in the evidence

Our confidence in the evidence for each outcome for each intervention will be assessed using an extension of the Grading of Recommendations Assessment, Development, and Evaluation framework developed for network meta-analysis and implemented in the CINeMA tool.[53 54]

### Systematic review 2: UKR and TKR

The systematic review and meta-analysis methods used comparing UKR with TKR in review 2 will be similar to those used in review 1.

### Eligibility criteria

*Types of studies*

We will include RCTs.

*Participants*

Eligible patients will be as described in review 1 but will have been assessed as eligible for both an elective UKR and an elective TKR.

*Interventions and comparators*

The knee replacement constructs will include all components used in medial, lateral and patellofemoral UKR and in TKR.

To complete the network of studies in review 2, we will include studies comparing:

(2a) different types of UKR constructs.

(2b) UKR with TKR constructs.

(2 c) TKR constructs for patients eligible for UKR.

Ascertaining the eligibility of trial participants for 2c will be challenging due to paucity of information on patient characteristics described at trial level (eg, number of symptomatic compartments) and time trends in clinical practice. Clinicians in our team will assess which trials may 'possibly include' patients potentially eligible for UKR in a gradient up to studies that 'definitely exclude' patients eligible for UKR.

*Types of outcome measures*

Outcome measures will be as described in review 1.

### Search strategy

The search in review 1 includes terms for UKR and thus will identify relevant studies. Methods used will mirror those of review 1.

### Data management

Management of study information, screening and data extraction will be as described in review 1. Data extraction will additionally include details of whether the UKR is for treatment of osteoarthritis in the medial, lateral or patellofemoral compartment.

### Risk of bias assessment

Assessment of risk of bias will use methods described in review 1.

### Data analysis

For pairwise and network meta-analyses, we plan main analyses restricted to studies (2a) comparing different UKR constructs and (2b) comparing UKR and TKR constructs. Secondary analyses (2 c) will also include comparisons of different TKR constructs in patients eligible for UKR.

For review 2a, the reference construct will be the most widely used cemented mobile bearing. In reviews 2b and 2 c, the reference construct within the network meta-analysis will be the TKR combination of cemented, cruciate-retaining, fixed-bearing implant components with metal on polyethylene bearing materials.

The methods for analysis will otherwise be as in review 1.

### Joint registry analysis

#### Joint registries

In our previous research,[55] we have established close links between the National Joint Registry for England, Wales, Northern Ireland and the Isle of Man (NJR) and the Swedish Hip Arthroplasty Register.[56] This was a successful collaboration, and our analyses benefited from the use of similar data collection and management in the two databases and the long follow-up of patients in Sweden. Recognising this, we have established links with the SKAR.[57]

Data collection in the NJR commenced on 1 April 2003 and includes 1 193 830 primary knee replacements with verifiable patient data up to 31 December 2018.[1] In 2020, up to 17 years of patient follow-up data are available for analysis.

SKAR was established in 1975 and includes patient identifier numbers that allows tracking of other healthcare use.[58] More complete data with information required for our study has been collected since 1989. Thus, in 2020, the registry includes up to 30 years of patient follow-up data.

### Eligibility criteria

*Patient group*

Patients included in analyses will be aged 18 years or over, undergoing primary TKR or UKR for osteoarthritis, with valid data on age, sex and construct characteristics.

## Interventions and comparisons

Knee replacement constructs for comparison will be characterised by bearing mobility, constraint, fixation and bearing materials. Patella resurfacing is recorded in the NJR but is infrequently used in Sweden. We will exclude patients receiving patellofemoral and bicompartmental knee replacement as they are generally used in a selected patient group.

The reference construct will be TKR with cemented, cruciate-retaining, fixed-bearing implant components with metal on polyethylene bearing materials.

## Outcomes

Outcomes for analyses will be time to first, second and subsequent revision surgeries and death after primary surgery.

## Statistical analysis

We will prepare a joint statistical analysis plan for analysis of NJR and SKAR data for our research questions. The analysis plan will include: definitions of primary surgery, knee replacement constructs and techniques, follow-up time frames and age groups; statistical methods including handling of missing data; and patient characteristics for a possible subgroup of TKR patients eligible for UKR. We will estimate hazard rates of first and second revision, HRs between implants and the effect of covariates (eg, age, gender, time since primary surgery and time since first revision) on these parameters.

The analyses will be performed using STATA 15.1 software. Estimates from the SKAR population will be calibrated to the NJR population using a range of calibration models implemented in OpenBUGs software as in our previous study.[55] These calibrations will model some difference between SKAR and NJR estimates of hazard rates and HRs that is either fixed, random or independent over time periods.

## Cost-effectiveness analysis

### Economic decision models

RCT and registry data will be combined with cost data and utility weight estimates to inform a probabilistic lifetime decision economic model to determine the relative cost-effectiveness from an NHS and social services perspective of: (1) TKR constructs and (2) TKR and UKR constructs for patients eligible for both. Results will be stratified by sex and age group. We will limit knee replacement constructs to those available for clinical use in the NHS.

We will calculate expected costs and quality-adjusted life years (QALYs), discounted at 3.5% per annum using a fully probabilistic analysis, which reflects parameter uncertainty in the sampled distributions, and simulating some number of iterations. We will first use 10 000 iterations and check if it is sufficient for the mean and SD of the cost and QALYs to converge; more simulations will be employed if needed. The reference construct for both models will be TKR with cemented, cruciate-retaining,

fixed-bearing implant components with metal on polyethylene-bearing materials.

The economic models will be implemented in the open source statistical software R.[59] The R software has well-documented advantages of speed, flexibility and transparency over other software such as Excel.[60] The same models will be fitted to both decision populations. Two model structures will be explored (full details are included in online supplemental material). First is a cohort Markov multistate model with health states representing time since primary surgery, post first revision, post second or higher revision and death.[61] States for time since primary will correspond to <3 years, ≥3 years but <10 years and ≥10 years postprimary. The post first revision states will correspond to early (<3 years), middle (≥3 years but <10 years) and late (≥10 years) revision. Probabilities of first revision will depend on the time since primary, while probabilities of second revision will depend on whether the first revision was early, middle or late. Probabilities of first revision for each construct and time period and for second revision will be calculated using area under the curve of estimated hazard rates.[62] For first revision, rates for the reference construct will come from the NJR/SKAR analysis. Rates of first revision for other constructs will be calculated by applying time-period specific HRs from the network meta-analysis or, if RCT and thus network meta-analysis data are not available, NJR/SKAR data.

The second model structure we will use is an individual-level continuous time semi-Markov model.[63] We aim to employ the 'hesim' extension to R to implement this model.[64] States will correspond to no revision, post first revision, post second revision and death. Transition rates of first revision will depend on time since primary surgery and patient characteristics (eg, age and gender). Rates of second revision will depend on time from primary to first revision and time since first revision. Model parameters will be estimated using the same data as for the cohort Markov model but without a conversion to probabilities. NJR/SKAR will provide rates of first revision for our reference construct. These will be adjusted to other constructs using HRs from the network meta-analysis or, if RCT and thus network meta-analysis data are not available for the construct or time period, NJR/SKAR estimates of HRs. NJR/SKAR will provide rates of second revision, which will be assumed common across constructs. This is a refinement of the cohort Markov multistate model and captures the main deviations from a Markov model in the disease area. While results are expected to be similar, our base case will be the individual-level continuous time semi-Markov model as it places fewer restrictions on timings of events.

We will conduct a literature search to identify the best data source to derive utility weight estimates stratified by age and sex for use in the models. A potential source is the UK Patient Reported Outcome Measures (PROMs) dataset for patients with knee replacement.[65] The PROMs dataset includes EuroQol Questionnaire (EQ-5D-3L) data at 6 months postsurgery for patients over 40 years old.

Construct prices in GB pounds will be estimated from a national database of NHS trusts providing implant procurement prices to the NJR: the NJR INFORM Implant price-benchmarking database.[66] Primary and revision surgeries, and follow-up healthcare costs will be obtained from national tariffs.[67] We will search the literature for published estimates of primary and secondary care follow-up costs from cost-effectiveness analyses in RCTs of knee replacement. The validity of these costs in the context of our implant construct comparisons will be assessed by clinicians in our team.

Cost-effectiveness will be estimated using the mean incremental net monetary benefit statistic (INMB) for each knee replacement construct compared with the reference construct, a willingness-to-pay threshold of £20000 per QALY.[68] The construct with the highest INMB is the most cost-effective for each patient subgroup. With cost-effectiveness acceptability curves, we will show how the probability of knee replacement constructs being most cost-effective varies as willingness-to-pay thresholds change.

In sensitivity analyses we will assess the robustness of the results to changes in key parameters and assumptions. These will involve variation of: time periods over which event hazards are assumed constant; estimations of transition probabilities (network meta-analysis only, registries only and combination of both); model structure to include different health states; costs of constructs; time in theatre to perform different surgeries; revision TKR costs; and other follow-up costs.

## ETHICS AND DISSEMINATION

The systematic reviews will not require ethical approval as we are undertaking secondary analyses of published data. For the analyses of joint registry data, we will design a common analysis plan and provide representatives of the NJR and SKAR with information required for submission to their respective ethics committees. The project has been assessed with the NHS REC committee and does not require its ethical review.

We expect the results from our study to provide direct patient benefit through better information available for clinical surgical teams performing TKRs, which will influence the commissioning of knee replacement constructs used at NHS trusts in the UK. We will publish our effectiveness and cost-effectiveness findings in two separate peer-reviewed journal articles in high-impact open-access journals. The findings of our studies will also be disseminated to the NHS, NJR, clinicians, societies, charities, patients and the general public via presentations, reports and websites.

## DISCUSSION

Surgeons and patients are faced with many construct and technique options for use in knee replacement. Patients in our PPI group told us that they were not informed about surgical choices when discussing surgery with their consultant. In a stakeholders meeting, knee surgeons advised us that they chose knee replacement constructs based on their preferences, surgical skills, availability of implants from manufacturers and available research evidence. There is a need for a comprehensive synthesis of evidence to inform this choice.

Given the large number of knee replacements performed annually and stringent NHS budgets, it is important to determine which knee replacement constructs provide the best outcomes for patients at lowest cost to the NHS. Ideally, all constructs would be compared in an RCT with long-term follow-up and enough statistical power to estimate differences in revision rates between constructs. However, it would be difficult to compare large numbers of constructs in an RCT and fund long-term follow-up.

We suggest that a more practical and efficient approach is to use all available evidence in published head-to-head RCTs to obtain treatment effect estimates for knee replacement constructs. We will complement these with analyses of large longitudinal national registries of surgical and patient data with long-term follow-up. Both estimates will be used to produce economic decision models that will rank and estimate the cost-effectiveness of knee replacement constructs.

**Author affiliations**
[1]Musculoskeletal Research Unit, Bristol Medical School, Bristol, UK
[2]Department of Population Health Sciences, Bristol Medical School, Bristol, UK
[3]National Institute for Health Research Bristol Biomedical Research Centre, University of Bristol, Bristol, UK

**Contributors** All authors contributed to the concept and design of the study. EMRM, HT and ADB drafted the article, and JD, JH, NW, AB, LH, MRW and AWB revised it critically for important intellectual content. EMRM and ADB take responsibility for the integrity of the work as a whole, from inception to finished article.

**Funding** This study is funded by the National Institute for Health Research (NIHR) Research for Patient Benefit programme (project reference PB-PG-1013–32010).

**Disclaimer** The views expressed are those of the authors and not necessarily those of the NIHR or the Department of Health and Social Care.

**Competing interests** None declared.

**Patient consent for publication** Not required.

**Provenance and peer review** Not commissioned; externally peer reviewed.

**ORCID iDs**
Elsa M R Marques http://orcid.org/0000-0003-1360-5677
Jane Dennis http://orcid.org/0000-0001-9718-2653
Andrew D Beswick http://orcid.org/0000-0002-7032-7514
Michael R Whitehouse http://orcid.org/0000-0003-2436-9024

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
