## [Reviewer comments · BMJ Open]

ARTICLE DETAILS

TITLE (PROVISIONAL)	The choice between implants in knee replacement: Protocol for a Bayesian network meta-analysis, analysis of joint registries and economic decision model to determine the effectiveness and cost-effectiveness of knee implants for NHS patients. The KNeE Implant Prosthesis Study (KNIPS)
AUTHORS	Marques, Elsa; Dennis, Jane; Beswick, Andrew; Higgins, Julian; Thom, H; Welton, Nicky; Burston, Amanda; Hunt, Linda; Whitehouse, Michael; Blom, AW

VERSION 1 – REVIEW

REVIEWER	Mika Niemeläinen Coxa the Hospital for Joint Replacements, Finland
REVIEW RETURNED	01-Jun-2020

GENERAL COMMENTS	I thank this opportunity to review these study protocols. Subjects in general are interesting and study questions are relevant for future analysis. In TKR construct options monobloc is mentioned as one-piece poly implant. There is also other monoblocks like TM monoblock tibias which may be included in the analysis. At least SKAR includes these implant. The role of the SKAR is not clear for me because none of the authors in protocol study group are from the SKAR. Should the SKAR data be used in sensitivity analysis or maybe some authors for example from Lund should be included? This may cause some bias at reporting phase.
---

REVIEWER	Paul Baker South Tees NHS Foundation Trust
REVIEW RETURNED	07-Jun-2020

GENERAL COMMENTS	My only concern is the study's ability to answer objective 2. However, this is a methodological concern regarding the data available to answer this objective and should not stop the protocol from being published. Regarding objective 2, NICE recently undertook an analyses of all RCTs and associated cost effectiveness analyses for medial UKR versus TKR as part of its guideline process. While this study expands to include all RCTS for all UKR types (including medial, lateral and PF UKR types) I worry, that given the limited level 1 evidence available for medial UKR, that there will not be enough RCT evidence to allow a robust NMA to be constructed given the
--

	paucity of comparative trials including a UKR arm. There is a risk of the analysis being skewed by the results of 1 large RCT (e.g. TOPKAT). I also cannot see how registry data can be used for this objective as you cannot define who might be eligible for both UKR and TKR based on the demographic / surgical data collected within registries. The decision for UKR versus TKR is complex and based on a range of patient, disease, radiographic and clinical findings. Registry data cannot adequately determine eligibility and as such any analysis and conclusions drawn from this data will be heavily caveated and potentially flawed.
--	--

REVIEWER	Ove Furnes Haukeland university hospital, Bergen, Norway
REVIEW RETURNED	09-Jun-2020

GENERAL COMMENTS	The choice between implants in knee replacement: Protocol for Bayesian network meta-analysis, analysis of joint registries and economic decision model to determine the effectiveness and cost-effectiveness of knee implants for NHS patients. The KNess Implant Prostheses Study (KNIPS). This is a clinical relevant study that will help clinicians to choose the best available implants for their knee osteoarthritis patients. The methodology seems appropriate. The only issue I have with the study is table 1 which outline the TKR construct options. 1. Fixation: An emerging fixation strategy is inverse hybrid where the tibial component is uncemented and the femoral component cemented. I recommend this to be included (Niemeläinen MJ, Mäkelä KT, Robertsson O, W-Dahl A, Furnes O, Fenstad AM, Pedersen AB, Schrøder HM, Reito A, Eskelinen A. The effect of fixation type on the survivorship of contemporary total knee arthroplasty in patients younger than 65 years of age: a register-based study of 115,177 knees in the Nordic arthroplasty register association (NARA) 2000-2016. Acta Orthop. 2020 Apr;91(2):184-190.) 2. Constraint and identification of implants. The classification in the registries and RCTs can be difficult to compare. Do the study have a plan for how to do this in a proper manner. The ISAR library can be used, using Catalogue (REF-numbers) to identify the implants and compare that they are the same as has been done in the study by Gøthesen et al. (Gøthesen Ø, Lygre SHL, Lorimer M, Graves S, Furnes O. Increased risk of aseptic loosening for 43525 rotating-platform vs. fixed-bearing total knee replacements. Acta Orthop 2017; Dec;88(6):649-656). An example is the NexGen knee, which is one of the most used TKR, there are several cemented models of the tibia like Option stemmed cemented and Precoated stemmed cemented which have the same geometry, but the precoated version has PMMA coating. These are different implants with different price and should be differentiated. How can this be done? 3. Bearing material. Some tibial implants are monoblock but not all poly, such as the AGC implant as described in Gøthesen et al (Gøthesen Ø, Espehaug B, Havelin L, Petursson G, Lygre SH, Ellison P, Hallan G, Furnes O. Survival rates and causes of revision in cemented primary total knee replacement. A report from the Norwegian Arthroplasty Register 1994-2009. Bone Joint J 2013;95-B:636-42). This applies both for TKR and for UKR.
---

	4. Patella should have a fixation option
--	--

REVIEWER	Josie Athens University of Otago
REVIEW RETURNED	18-Aug-2020

GENERAL COMMENTS	The authors present a well written and supported protocol for a meta-analysis on the choice between knee implants. The KNIPS will provide valued information, though the authors mentioned that, on a first glance, options for knee implants were not discussed with patients; this finding makes hard to evaluate the actual contribution of the final study. Regarding the proposed methodology, the authors propose the hazard rate ratio as the statistic to compare different combinations of knee implants. Given that:  1. The primary outcome of interest will be the knee implant revision rate. 2. A limitation of the study is the likely inclusion of RCTs with short follow up. 3. Failure of implants is expected to follow an exponential distribution. 4. They assume hazard is constant. I propose the use of parametric methods (survival time ratios) instead of semi-parametric ones (hazard rate ratios). From these models, predicting the time of implant failure (even outside followed-up time) would be possible and of interest. The economic decision models are out of my reach of expertise, so I cannot comment on those. I suggest for the authors to be more clear on the methods, in particular, why they propose a semi-Markov model over, for example, ordinary differential equations for the continuous-time alternative. In summary, it is not clear to me what is the justification for publishing a protocol for a meta-analysis. In this particular case, if this is to present a novel theoretical approach, more discussion should be put on describing the models.
--

REVIEWER	CAI Jingheng Sun Yat-sen University P.R.CHINA
REVIEW RETURNED	28-Sep-2020

GENERAL COMMENTS	I have one concern about the proposed methods and software.  ● Page 12, lines 36—39. The implementations of network meta-analysis using OpenBUGs software need to be presented in details.
--

REVIEWER	Christopher N. Graham, MS RTI-Health Solutions, Research Triangle Park, NC, USA
REVIEW RETURNED	02-Oct-2020

GENERAL COMMENTS	Overall, this appears to be a well designed protocol. There are a few
---

pieces that need clarification and further detail.

Abstract

- Page 5 (as listed in the top left of each page of the version provided by the journal, not the authors' page number at the bottom right), lines 16 and 19: Please specify that "survival" is the survival of the joint and not patient survival (death)

Manuscript

- Page 7, line 6: You need to define UK (United Kingdom). Also, it seems odd to also cite statistics from two countries in the first sentence with Sweden being one. While a tremendous country, Sweden is not often accentuated this early in manuscripts. Are there other statistics that you could mention to make this more global (other counties), or is there a way to explain why Sweden is cited this early?

- Page 7, line 25: Need to link osteoarthritis and knee replacement with a sentence prior to osteoarthritis's first mention.

- Page 7, line 36: Clarify what you mean by "has a shorter duration". Is that time in surgery or duration/life of the replacement?

- Page 7, line 44: Many surgeons prefer TKR, but then you cite the rates for UKR in the next couple of sentences. It would read better if the rates cited were for TKR, or if you added TKR in addition to UKR.

- Page 8, line 12: You mention "search of the same databases" but you never list which databases were searched in the first place. Add the list of databases searched to page 8, line 48.

- Page 11, line 49: UKR is mentioned here in the TKR section. Presumably, this is because the TKR vs. UKR section utilizes many of the same methods as the TKR section and you did not want to repeat yourselves. However, to be accurate, this section should only mention TKR first revision and not the UKR. Add detail to the TKR vs. UKR section if needed. Also, it is not fully clear to me why two separate systematic reviews are needed. Presumably, you will have a ton of overlap in article hits between the two. Are you planning on doing the screens twice and exclusions twice? If so, that is fine. However, if you are planning to do one big lit. search, remove duplicates, etc. and then partitioned the articles into TKR only and TRK vs. UKR groups for data extraction then I think the text should be revised to reflect this (i.e., it would be 1 systematic lit. review and not 2). If the analysis are different between the two, that can be split into two sections if needed.

- Page 11, line 52: You discuss extracting data from published KM "curves". These are not "curves" per se as they are stepped. I would prefer "plots" rather than "curves" be used. Additionally, when people usually talk about extracting data from KM plots to recreate patient-level data via Guyot method (numbers at risk, drop points over time) they are doing so for survival analyses. However, when you discuss the data extraction on page 13 (lines 12-22) you only list things like means, medians, IQR, etc. Are you planning to conduct survival analyses? If so, then detail needs to be added. If not, then I suggest clarifying what exactly you plan to extract from the KM plots (e.g., reading medians off plots if not directly reported in text).

- Page 13, line 38: Can you please add a sentence or two specify exactly how levels of bias will be accessed? As high-risk of bias studies will be excluded, it will be important to be transparent on what exactly is a "high-risk of bias study".

- Page 14, line 7: Define exactly what "standard meta-analysis" you are going to employ. Are you talking about frequentist random and fixed effects models? Add specifically what you mean by "standard".

	- Page 14, line 18: Through your prior searches were you able to ascertain whether it will be feasible to connect the network? If so, that would be relevant information to include in the preliminary literature search section (i.e., a feasibility assessment was conducted). Otherwise, you are planning for a lot of work that may not be possible. Additionally, if a feasibility assessment was not conducted, then you need to describe alternate methods for comparison and their limitations if it is not possible to connect the network. - Page 17, line 12: This is where Sweden comes in, which is fine. I just think it is odd to have it in the very first sentence of the manuscript without any context. - Page 18, line 28: You are not using “QALY data”. The quality-adjusted life year is a result of time in a health state x a utility weight. You are using utility weight estimates in the model. Please revise. Line 36 is correct; it is line 28 that you need to change. Also, you need to define what a QALY is prior to using the abbreviation. - Page 18, line 48 – Are you planning to use any specific R packages with the cohort Markov like you are hesim for the second model? If so, then define here. There also needs to be more rationale and justification for building two models. What is the purpose of doing this? If one is better than the other, then why not just do one? Also, if there is a rationale for two, then the base-case should come first (hesim model) and not second. - Page 18, line 59: Spelling issue with “earl”. - Page 19, line 23: Again, it’s not QALYs here but utility weights. Be specific and accurate. - Page 18, line 27: It’s great to have the EQ-5D here, but the applicability of utility weights from a single point in time (6 months post-surgery) to health states that have other time-based components is suspect. Also, it is limited to 40 years and older and you’re talking about stratifying by age. - Page 18, line 35: Costs based on prior CE models need to be reviewed with clinical experts to make sure they are logical and truly represent the costs born by patients. Suggest adding in a clinical review for face validity for these costs. - Model parameters overall: You have not adequately described how you plan to incorporate time components into the analysis. You mention hazard ratios in the NMA section, which is fine to show the relative effect between interventions, but what are those hazard ratios being applied to within the model. This has to be added as well as an assumption if time movement is a constant rate (exponential via conversion from mean or median estimates), or if you plan have time-dependent probabilities via parametric survival analyses? This piece needs to be added prior to publication.
--	--

VERSION 1 – AUTHOR RESPONSE

Reviewer 1

I thank this opportunity to review these study protocols. Subjects in general are interesting and study questions are relevant for future analysis.

Response: Thank you for this encouragement.

In TKR construct options monobloc is mentioned as one-piece poly implant. There is also other monoblocks like TM monoblock tibias which may be included in the analysis. At least SKAR includes these implant.

Response: Thank you. We have now noted that monobloc implants may be polyethylene or metal with polyethylene and this will be considered in the network meta-analysis nodes. In registry analyses, we anticipate that we may group all monoblocs, despite potentially different constructs, under one monobloc type due to sparsity of data for analysis. We will however aim to keep the groups distinct where numbers allow.

The role of the SKAR is not clear for me because none of the authors in protocol study group are from the SKAR. Should the SKAR data be used in sensitivity analysis or maybe some authors for example from Lund should be included? This may cause some bias at reporting phase.

Response: We have an agreement with SKAR to analyse registry data. This is a valuable resource for our research as the registry was established earlier than the NJR and follow up time is longer, an important consideration when looking at implant survival. Both the NJR and SKAR will provide estimates of revision rates from their registries using a common statistical analysis plan. SKAR estimates will be calibrated to the UK population for the economic model where NJR lacks data. A similar approach has been used successfully in our previous work in implants for hip replacement (Fawsitt et al. Choice of prosthetic implant combinations in total hip replacement: Cost-effectiveness analysis using UK and Swedish hip joint registries data. Value Health. 2018;22:303-12 PubMed).

Reviewer 2.

My only concern is the study's ability to answer objective 2. However, this is a methodological concern regarding the data available to answer this objective and should not stop the protocol from being published.

Regarding objective 2, NICE recently undertook an analyses of all RCTs and associated cost effectiveness analyses for medial UKR versus TKR as part of its guideline process. While this study expands to include all RCTS for all UKR types (including medial, lateral and PF UKR types) I worry, that given the limited level 1 evidence available for medial UKR, that there will not be enough RCT evidence to allow a robust NMA to be constructed given the paucity of comparative trials including a UKR arm. There is a risk of the analysis being skewed by the results of 1 large RCT (e.g. TOPKAT). I also cannot see how registry data can be used for this objective as you cannot define who might be eligible for both UKR and TKR based on the demographic / surgical data collected within registries. The decision for UKR versus TKR is complex and based on a range of patient, disease, radiographic and clinical findings. Registry data cannot adequately determine eligibility and as such any analysis and conclusions drawn from this data will be heavily caveated and potentially flawed.

The reviewer raises important issues that we are concerned about as well. Our review is different from previous reviews, as we are trying to overcome some of their limitations. We will use only RCTs and these will be assessed for risk of bias using the Cochrane RoB2 tool. It is frequently the case that results are “skewed” by larger trials, but we can also employ statistical methods (e.g. random effects vs fixed effects models) to account for these. In addition, whether it will be possible to build an economic model on the NMA alone or not, we will always use NJR data in conjunction with or instead of the NMA results, at least in sensitivity analyses.

We also agree that previously published results based on analysis of NJR data have not accurately portrayed the eligibility for TKR that surgeons employ in their clinical practice. The decision to offer a TKR or UKR to a patient may rely on factors such as the distribution of osteoarthritis in the joint, where the patient is experiencing pain, and preferences for recovery time and functional demands. There are no data collected on such factors in registries or trials, so we need to rely on proxy variables to control for these factors. Our co-authors MRW and LPH have described a different statistical method (Hunt et al. 2020. Patients receiving a primary unicompartmental knee replacement have a higher risk of revision but a lower risk of mortality than predicted had they received a total knee replacement: data from the National Joint Registry for England, Wales, Northern Ireland, and the Isle of Man. J Arthroplasty; doi: 10.1016/j.arth.2020.08.063.). This developed a predictive model using Flexible Parametric Survival modelling to accommodate the time varying effects of the predictor

variables. This will allow us to model what the revision rates would have been if patients who received UKR had TKR, based on predictor variables such as age, sex, year of primary and BMI. Their findings are different from those using propensity score matching, and we believe they may be closer, albeit still imperfect, to the reality of clinical practice.

Our findings will be subjected to a range of sensitivity analyses. We will run models using estimates from trials only, from registries only (using a range of calibration methods), and combining the two sources of data, which is the strongest and most likely base-case scenario. All uncertainty, strengths and limitations will be clearly explained when interpreting the results for clear decision-making.

Reviewer 3

This is a clinical relevant study that will help clinicians to choose the best available implants for their knee osteoarthritis patients. The methodology seems appropriate.

The only issue I have with the study is table 1 which outline the TKR construct options.

1. Fixation: An emerging fixation strategy is inverse hybrid where the tibial component is uncemented and the femoral component cemented. I recommend this to be included (Niemeläinen MJ, Mäkelä KT, Robertsson O, W-Dahl A, Furnes O, Fenstad AM, Pedersen AB, Schrøder HM, Reito A, Eskelinen A. The effect of fixation type on the survivorship of contemporary total knee arthroplasty in patients younger than 65 years of age: a register-based study of 115,177 knees in the Nordic arthroplasty register association (NARA) 2000-2016. Acta Orthop. 2020 Apr;91(2):184-190.)

Response: In UK joint registries, hybrid and inverse hybrid knee replacements are reported together as few of the latter are undertaken. As yet, we believe no randomised evaluations are reported. Recognising that inverse hybrid fixation is of particular interest in Sweden where it has been used more than hybrid in at least one TKR system, we now note this in “TKR construct options” paragraph 4 (with citation) and in table 1, and will aim to include this in our SKAR and NJR analysis request.

2. Constraint and identification of implants. The classification in the registries and RCTs can be difficult to compare. Do the study have a plan for how to do this in a proper manner. The ISAR library can be used, using Catalogue (REF-numbers) to identify the implants and compare that they are the same as has been done in the study by Gøthesen et al.(Gøthesen Ø, Lygre SHL, Lorimer M, Graves S, Furnes O. Increased risk of aseptic loosening for 43525 rotating-platform vs. fixed-bearing total knee replacements. Acta Orthop 2017; Dec;88(6):649-656). An example is the NexGen knee, which is one of the most used TKR, there are several cemented models of the tibia like Option stemmed cemented and Precoated stemmed cemented which have the same geometry, but the precoated version has PMMA coating. These are different implants with different price and should be differentiated. How can this be done?

Response: We will be differentiating types of implants in all of our analyses: the review of the literature and the analysis of registry data, to inform our economic model. From RCTs, we will extract as much data as possible that describes the features of implants. From our clinical advisors in the team and stakeholder meetings, these were bearing mobility, constraint, materials, and fixation to the bone, as described in table 1, page 27 of the manuscript. We will attempt to make data more complete by considering the date of the study and the characteristics of the systems at the time – e.g. type of polyethylene or coatings if possible. Many articles have representative x-ray pictures which will be checked. Up to the end of 2016, 449 different knee replacement constructs used in over 500 operations were recorded in the NJR for England, Wales, Northern Ireland, and the Isle of Man (Deere et al. 2019). It would be impossible to rank them individually and we have to make decisions in bulking them together under a minimum set of characteristics (suggested in table 1) that would be detailed enough to pick up variation in important outcomes. We will use the most up-to-date procurement prices to cost different implant types. Different brands will have different prices and will be contributing to an “average” implant cost per implant type. We will perform sensitivity analyses on the implant prices (use most expensive within a type, using the cheapest within a type, etc) to check

the robustness of our results. For our results to be generalisable, we would like to make our implant prices as realistic and average as possible to reflect real procurement decisions at the hospital level.

3. Bearing material. Some tibial implants are monoblock but not all poly, such as the AGC implant as described in Gøthesen et al (Gøthesen Ø, Espehaug B, Havelin L, Petursson G, Lygre SH, Ellison P, Hallan G, Furnes O. Survival rates and causes of revision in cemented primary total knee replacement. A report from the Norwegian Arthroplasty Register 1994-2009. Bone Joint J 2013;95-B:636-42). This applies both for TKR and for UKR.

Response: Thank you for this. Reviewer 1 also noted this, and we have added details in the TKR construct options section and Table 1.

4. Patella should have a fixation option

Response: We have not yet identified randomised comparisons of cemented with cementless patella fixation. However, recognising that this will potentially be recorded in joint registries, we have added appropriate text in "TKR construct options" paragraph 4 and specified the options in Table 1.

Reviewer 4

The authors present a well written and supported protocol for a meta-analysis on the choice between knee implants.

The KNIPS will provide valued information, though the authors mentioned that, on a first glance, options for knee implants were not discussed with patients; this finding makes hard to evaluate the actual contribution of the final study.

Response: Thank you, this is an important point. A major anticipated contribution of the study is to inform surgeons of the most effective and cost-effective knee constructs for better informed choices for their patients. It will also raise awareness in patients of the myriad of choices available to them which will prompt them to discuss them with the surgeons. This raised awareness will also help them to better understand information about implants received at the surgeon appointment which they may have previously disregarded.

We have added to the end of the Patient and public involvement section on pages 8-9, "Anticipated contributions will include advice on dissemination of results, particularly plain language summaries and the use of our findings to support shared decision making."

Regarding the proposed methodology, the authors propose the hazard rate ratio as the statistic to compare different combinations of knee implants. Given that:

- 1. The primary outcome of interest will be the knee implant revision rate.**
- 2. A limitation of the study is the likely inclusion of RCTs with short follow up.**
- 3. Failure of implants is expected to follow an exponential distribution.**
- 4. They assume hazard is constant.**

I propose the use of parametric methods (survival time ratios) instead of semi-parametric ones (hazard rate ratios). From these models, predicting the time of implant failure (even outside followed-up time) would be possible and of interest.

Response: Within the RCT follow-up period, we assume a constant hazard rate within a period, but are splitting the full time frame in different periods. This is more flexible than a single hazard rate ratio. We will explore alternative methods (parametric models) in sensitivity analyses if the fit of our base case model is poor and describe this in the protocol.

We will be using the National Joint Registry (NJR) and Swedish Knee Arthroplasty Register (SKAR) for longer term implant failure rates. This is more reliable than using assumed parametric distributions to extrapolate outside of the RCT follow-up times, as recommended by the UK National Institute for

Health and Care Excellence (NICE) Decision Support Unit. We have added text and a citation to explain this choice in page 12 of the methods, network meta-analysis subheading.

“We do not aim to extrapolate outside of the follow-up time of RCTs as this would rely on assumptions about the parametric form; as explained below we will use registry data for long term rates, in line with recommendations of the NICE Decision Support Unit.”

The economic decision models are out of my reach of expertise, so I cannot comment on those. I suggest for the authors to be more clear on the methods, in particular, why they propose a semi-Markov model over, for example, ordinary differential equations for the continuous-time alternative.

Response: The semi-Markov approach captures the key non-Markov elements of the condition being modelled. These are the dependence on individual characteristics (age, gender), rates of 1st revision depending on time since primary surgery, and rates of 2nd revision depending on time from primary replacement to 1st revision. Going beyond this is not necessary. The semi-Markov method is also easily implemented in the R package ‘hesim’. To clarify, we have added to the final sentence on the modelling approach in page 17 of the “Economic decision models” methods:

“This is a refinement of the cohort Markov multistate model and captures the main deviations from a Markov model in the disease area. While results are expected to be similar, our base case will be the individual level continuous time semi-Markov model as it places fewer restrictions on timings of events.”

In summary, it is not clear to me what is the justification for publishing a protocol for a meta-analysis. In this particular case, if this is to present a novel theoretical approach, more discussion should be put on describing the models.

Response: While we have registered the systematic review protocol in PROSPERO, the main advantage of publishing our protocol in BMJ Open is to prospectively record our objectives, outcomes, and methods for transparency in the reporting of our results later on. In writing the protocol, we also hoped to add value by describing in detail knee implant options as well as complex methods for analyses. The peer reviews provided have helped us to be more inclusive (patella fixation, inverse hybrid fixation, description of methods) in these regards.

Reviewer 5

I have one concern about the proposed methods and software.

Page 12, lines 36—39. The implementations of network meta-analysis using OpenBUGS software need to be presented in details.

Response: Thank you for the opportunity to clarify this. The OpenBUGS code will be developed as part of our project. However, it will be based on the OpenBUGS code published in Lu et al, 2007, which we adapted in our hip replacement surgery work (Lopez-Lopez et al. 2017). We have added the following text to the Network meta-analysis methods section in page 12:

“We will adapt code developed for our hip replacement surgery network meta-analysis,(46) which itself was adapted from published code.(44)”

Reviewer 6

Overall, this appears to be a well designed protocol. There are a few pieces that need clarification and further detail.

Abstract

- **Page 5 (as listed in the top left of each page of the version provided by the journal, not the authors’ page number at the bottom right), lines 16 and 19: Please specify that "survival" is the survival of the joint and not patient survival (death)**

Response: Apologies for this, we now describe “construct survival”.

Manuscript

- **Page 7, line 6: You need to define UK (United Kingdom). Also, it seems odd to also cite statistics from two countries in the first sentence with Sweden being one. While a tremendous country, Sweden is not often accentuated this early in manuscripts. Are there other statistics that you could mention to make this more global (other countries), or is there a way to explain why Sweden is cited this early?**

Response: UK now defined. As noted at the beginning of our responses to the reviews, we hope that including this Swedish data in the introduction is acceptable. Sweden is the country with largest registry of patients in the world, after the UK, with the longest follow-up (over 30 years), even longer than the UK. Other countries with long-term registries are Norway and Iceland but they are considerably smaller in size. Australia has a large registry of TKR patients, but the follow-up is shorter than the UK. Other countries who perform many thousands of TKRs a year, such as the USA, do not have a comprehensive national registry of patients or release their data for analysis.

We have specified in paragraph 1 of the Introduction that Sweden is a collaborating country. We had hoped to include a representative of SKAR as an author but with the ongoing merger of Swedish hip and knee arthroplasty registers, this has not proved possible at this time.

- **Page 7, line 25: Need to link osteoarthritis and knee replacement with a sentence prior to osteoarthritis's first mention.**

Response: We have revised these sentences to make clear the link between osteoarthritis in knee compartments, pain and disability, and the need for total or unicompartamental knee replacement.

- **Page 7, line 36: Clarify what you mean by “has a shorter duration”. Is that time in surgery or duration/life of the replacement?**

Response: We have now made clear that is the duration of the operation.

- **Page 7, line 44: Many surgeons prefer TKR, but then you cite the rates for UKR in the next couple of sentences. It would read better if the rates cited were for TKR, or if you added TKR in addition to UKR.**

Response: Thank you. We have now made this more logical by adding in the rates for TKR.

- **Page 8, line 12: You mention “search of the same databases” but you never list which databases were searched in the first place. Add the list of databases searched to page 8, line 48.**

Response: Searches were in MEDLINE and Embase and this is now specified.

- **Page 11, line 49: UKR is mentioned here in the TKR section. Presumably, this is because the TKR vs. UKR section utilizes many of the same methods as the TKR section and you did not want to repeat yourselves. However, to be accurate, this section should only mention TKR first revision and not the UKR. Add detail to the TKR vs. UKR section if needed. Also, it is not fully clear to me why two separate systematic reviews are needed. Presumably, you will have a ton of overlap in article hits between the two. Are you planning on doing the screens twice and exclusions twice? If so, that is fine. However, if you are planning to do one big lit. search, remove duplicates, etc. and then partitioned the articles into TKR only and TRK vs. UKR groups for data extraction then I think the text should be revised to reflect this (i.e., it would be 1 systematic lit. review and not 2). If the analysis are different between the two, that can be split into two sections if needed.**

Response: We agree that “and UKR” is not required in this sentence and it is now removed. In the UKR Methods Types of outcome measures section we state, “Outcome measures will be as described in Review 1.”

The main reason for reporting as two reviews relates to funding which is in two distinct parts. The reviews will be written up separately. We would like to report them in one protocol as methods are common to each review but maintain a distinct identity for each. One search will be

used, but we anticipate that an update will be required for the UKR study which will be undertaken second.

- **Page 11, line 52: You discuss extracting data from published KM “curves”. These are not “curves” per se as they are stepped. I would prefer “plots” rather than “curves” be used. Additionally, when people usually talk about extracting data from KM plots to recreate patient-level data via Guyot method (numbers at risk, drop points over time) they are doing so for survival analyses. However, when you discuss the data extraction on page 13 (lines 12-22) you only list things like means, medians, IQR, etc. Are you planning to conduct survival analyses? If so, then detail needs to be added. If not, then I suggest clarifying what exactly you plan to extract from the KM plots (e.g., reading medians off plots if not directly reported in text).**

Response: In “Types of outcome measures” and “Data extraction” sections we have now made clear that data from Kaplan-Meier plots will only be extracted for the outcome of revision.

- **Page 13, line 38: Can you please add a sentence or two specify exactly how levels of bias will be assessed? As high-risk of bias studies will be excluded, it will be important to be transparent on what exactly is a “high-risk of bias study”.**

Response: We will use the Cochrane Risk of Bias tool 2 which assesses 5 domains and generates judgments of “low risk of bias,” “some concerns,” or “high risk of bias”. As our inclusion criteria specifies that studies must be randomised and recognising that blinding of delivery of interventions in the context of surgery are limited, key concerns will relate to deviations from intended interventions (e.g. large proportion of patients not receiving randomised allocation) and bias due to missing outcome data (e.g. high or unequal loss to follow up). Our first impressions are that there will be few studies at “high risk of bias” for revision outcome, but that others will have “some concerns”. There will probably be more studies at “high risk of bias” for long-term surgeon assessed scores and patient-reported outcomes which require attendance at clinics or response to questionnaire surveys.

We have added text to the “Risk of bias assessment” section to explain this. Proposed sensitivity analyses relating to concerns for risk of bias are included in the “Heterogeneity and subgroup analyses” section.

- **Page 14, line 7: Define exactly what “standard meta-analysis” you are going to employ. Are you talking about frequentist random and fixed effects models? Add specifically what you mean by “standard”.**

Response: We apologise for this inaccuracy. We have now amended this to “pairwise meta-analyses”.

- **Page 14, line 18: Through your prior searches were you able to ascertain whether it will be feasible to connect the network? If so, that would be relevant information to include in the preliminary literature search section (i.e., a feasibility assessment was conducted). Otherwise, you are planning for a lot of work that may not be possible. Additionally, if a feasibility assessment was not conducted, then you need to describe alternate methods for comparison and their limitations if it is not possible to connect the network.**

Response: Yes, we believe it will be feasible to conduct NMA. We are not certain to which level of detail the network will be laid – how detailed each “node” or intervention will be. This will depend on how many studies will be dropped from the review due to lack of data for outcomes of interest or high risk of bias. We may need to drop certain implant types due to lack of available studies that included them; or drop certain implant characteristics (e.g. bearing surface materials). In the protocol, we felt we should be as detailed as possible with all the methods we could employ in the best case scenario, where a detailed network of interventions defined by bearing material, mobility, constraint and fixation is possible, and will cut down if only fewer characteristics are possible. We have now summarised this and added it as a limitation to the “Strengths and limitations of the study” section in page 4:

“Some studies we identify may not report relevant outcomes or be at high risk of bias which will reduce the number of studies and knee replacement constructs that we can include in analyses.”

- **Page 17, line 12: This is where Sweden comes in, which is fine. I just think it is odd to have it in the very first sentence of the manuscript without any context.**

Response: Hopefully introducing the relevance of Sweden to the project in the Introduction section makes this clearer.

- **Page 18, line 28: You are not using “QALY data”. The quality-adjusted life year is a result of time in a health state x a utility weight. You are using utility weight estimates in the model. Please revise. Line 36 is correct; it is line 28 that you need to change. Also, you need to define what a QALY is prior to using the abbreviation.**

Response: Thank you, the changes have been made.

- **Page 18, line 48 – Are you planning to use any specific R packages with the cohort Markov like you are hesim for the second model? If so, then define here. There also needs to be more rationale and justification for building two models. What is the purpose of doing this? If one is better than the other, then why not just do one? Also, if there is a rationale for two, then the base-case should come first (hesim model) and not second.**

We are opting for the continuous model in the first instance which, in theory, will be the more detailed and superior model. If we do not have enough data to populate the continuous model to a high methodological standard, we will revert to using the discrete state Markov model. This model is less demanding on data requirements and is a more broadly used and recognisable model construct also yielding high-validity results.

- **Page 18, line 59: Spelling issue with “earl”.**

Response: This has been changed.

- **Page 19, line 23: Again, it’s not QALYs here but utility weights. Be specific and accurate.**

Response: Thank you, the change has been made.

- **Page 18, line 27: It’s great to have the EQ-5D here, but the applicability of utility weights from a single point in time (6 months post-surgery) to health states that have other time-based components is suspect. Also, it is limited to 40 years and older and you’re talking about stratifying by age.**

Response: In our experience from previous trials data, utility weights after TKR do not change much after 6 months post-surgery. We therefore feel comfortable to apply the same utility weights for the average quality of life index spent in period for that health state. Our age stratification will start at under 55 years old, and in 10 year brackets until 85 years and older to reflect the distribution of the population that receive knee replacement.

- **Page 18, line 35: Costs based on prior CE models need to be reviewed with clinical experts to make sure they are logical and truly represent the costs born by patients. Suggest adding in a clinical review for face validity for these costs.**

Response: Thank you, this is important. We will also review the literature for updated cost estimates in recent studies. Our group at Bristol are working on several studies which will be publishing primary and revision TKR costs in the future. We will use these for our models if appropriate. To ensure the validity of cost estimates, surgeons on our team with experience conducting knee replacements will assess their relevance. We have added details of this in page 18:

“We will search the literature for published estimates of primary and secondary care follow-up costs from cost-effectiveness analyses in RCTs of knee replacement. The validity of these costs in the context of our implant construct comparisons will be assessed by clinicians in our team.”

- **Model parameters overall: You have not adequately described how you plan to incorporate time components into the analysis. You mention hazard ratios in the NMA section, which is fine to show the relative effect between interventions, but what are those hazard ratios being applied to within the model. This has to be added as well as an assumption if time movement is a constant rate (exponential via conversion from mean or median estimates), or if you plan have time-dependent probabilities via parametric survival analyses? This piece needs to be added prior to publication.**

Response: We agree this was not well specified. We have added the following text to the paragraph explaining the cohort Markov model in page 17:

“Probabilities of 1st revision will depend on the time since primary while probabilities of 2nd revision will depend on whether the 1st revision was early, middle, or late. Probabilities of 1st for each implant and time period and for 2nd revision will be calculated using area-under-the-curve of estimated hazard rates.(62) For 1st revision, rates for the reference implant will come from the NJR/SKAR analysis. Rates of 1st revision for other implants will be calculated by applying time-period specific hazard ratios from the network meta-analysis or, if RCT and thus network meta-analysis data are not available, NJR/SKAR data.”

And the following to the paragraph explaining the semi-Markov model:

“Model parameters are estimated using the same data as for the cohort Markov model, but without a conversion to probabilities. NJR/SKAR will provide rates of 1st revision for our reference implant. These will be adjusted to other implants using hazard ratios from the network meta-analysis or, if RCT and thus network meta-analysis data are not available for the implant or time period, NJR/SKAR estimates of hazard ratios. NJR/SKAR will provide rates of 2nd revision, which will be assumed common across implants.”

VERSION 2 – REVIEW

REVIEWER	Niemeläinen M Coxa the Hospital for Joint Replacements, Finland.
REVIEW RETURNED	22-Nov-2020
GENERAL COMMENTS	I thank authors for their responses on my comments. I have no further comments.
REVIEWER	P Baker South Tees Hospitals NHS Trust, UK
REVIEW RETURNED	27-Nov-2020
GENERAL COMMENTS	Thank you for resubmitting this revised manuscript. It presents a clear account of the intended analysis. I have no additional concerns that need addressing at this stage
REVIEWER	Jingheng CAI Sun Yat-sen University
REVIEW RETURNED	21-Nov-2020
GENERAL COMMENTS	I have no further comments.